# Fast Degradation of Rhodamine B by In Situ H_2_O_2_ Fenton System with Co and N Co-Doped Carbon Nanotubes

**DOI:** 10.3390/ma16072606

**Published:** 2023-03-24

**Authors:** Wei Cui, Jiahui Fang, Yuanyuan Wan, Xueyu Tao, Litong Guo, Qiyan Feng

**Affiliations:** 1School of Environment and Spatial Informatics, China University of Mining and Technology, Xuzhou 221116, China; 2School of Materials and Spatial Informatics, China University of Mining and Technology, Xuzhou 221116, China

**Keywords:** carbon nanotubes, E-Fenton, rhodamine B, degradation, mechanism

## Abstract

In this study, an E-fenton oxidation system based on Co-N co-doped carbon nanotubes (Co-N-CNTs) was designed. The Co-N-CNTs system showed fast degradation efficiency and reusability for the degradation of rhodamine B (RhB). The XRD and SEM results showed that the Co-N co-doped carbon nanotubes with diameters ranging from 40 to 400 nm were successfully prepared. The E-Fenton degradation performance of Co-N-CNTs was investigated via CV, LSV and AC impedance spectroscopy. The yield of H_2_O_2_ could reach 80 mg/L/h within 60 min, and the optimal voltage and preparation temperature for H_2_O_2_ yield in this system was −0.7 V (vs. SCE) and 800 °C. For the target pollutant of RhB, the fast removal of RhB was obtained via the Co-N-CNTS/E-Fenton system (about 91% RhB degradation occurred during 60 min), and the •OH played a major role in the RhB degradation. When the Fe^2+^ concentrations increased from 0.3 to 0.4 mM, the RhB degradation efficiency decreased from 91% to about 87%. The valence state of Co in the Co-N-C catalyst drove a Co^2+^/Co^3+^ cycle, which ensured the catalyst had good E-Fenton degradation efficiency. This work provides new insight into the mechanism of an E-Fenton system with carbon-based catalysts for the efficient degradation of RhB.

## 1. Introduction

The electro-Fenton process based on hydroxyl radicals has attracted widespread attention due to its environmental friendliness, high efficiency and energy-saving properties [1,2], during which, H_2_O_2_ is formed in situ by the reaction of the two-electron reduction of O_2_ (2e-ORR) on the cathode [3,4]. Then, the in situ-produced H_2_O_2_ reacts with the metal catalyst to produce radical •OH. The potential explosion hazard in the process of H_2_O_2_ production and transportation could be avoided by the on-site electrosynthesis of H_2_O_2_ during the electro-Fenton process [5]. Lower energy consumption for organic pollutants’ elimination could be obtained by higher H_2_O_2_ selectivity and regeneration rates of Fe^2+^, which are also the two key factors to promote the E-Fenton application [6,7].

Although excellent ORR catalytic activity and high H_2_O_2_ selectivity have been achieved by using noble-metal-based catalysts [8], their practical application in wastewater treatment was hindered by their scarcity and high cost [9].

Alternatively, carbon materials, especially biochar catalysts, have attracted increasing attention owing to their abundance, environmental sustainability, lost-cost and ease of fabrication [10]. In addition, carbon-based catalysts have attracted more attention due to their environmental sustainability, ease of manufacturing, low cost and abundance [11,12]. Moreover, the surface chemistry and pore structure of carbon-based catalysts can be easily adjusted to obtain more accessible active sites and faster mass transfer according to the preparation conditions [13,14]. The doping of metal, nitrogen and oxygen functional groups has been proved to be an effective strategy to improve their catalytic activity and H_2_O_2_ selectivity, which will significantly affect the oxidation performance. Co-based catalysts embedded in carbon matrixes, which have large specific surface areas, can effectively improve the catalytic activity and prevent Co from leaching [8,15]. The removal mechanism of rhodamine B via adsorption and catalysis has been investigated by many researchers [16,17,18,19]. Carbon nanotubes and graphene are typical carbonaceous materials [20,21,22], which are widely used for the removal of pollutants via catalysis and adsorption due to their large surface area and good Fenton efficiency.

Herein, a convenient and quick pyrolyzing process was used to prepare Co-N co-doped carbon nanotubes (Co-N-CNTs) as catalysts for the E-Fenton process. Electrochemical analyses were used to explore the degradation mechanism of RhB via E-Fenton [12]. The effects of different reaction conditions on RhB degradation were also investigated, including different preparation temperatures, Fe^2+^ concentrations and voltages [23].

## 2. Material and Methods

### 2.1. Synthesis of Co-N-C Composite Catalyst and Characterization

Some chemicals were purchased from Aladdin chemistry Co. Ltd. (Shanghai, China), such as rhodamine B (RhB, AR, ≥95.0%), and 5,5-Dimethyl-1-pyrroline N-oxide (DMPO, AR, ≥97.0%). Urea, Co(NO_3_)_2_·6H_2_O (AR, ≥98.5%), sulfuric acid (H_2_SO_4_, AR, 95.0~98.0%) and sodium hydroxide (NaOH, AR, ≥96.0%) were obtained from Sinopharm Chemical Reagent Co., Ltd. (Shanghai, China).

The Co-N-C composite catalysts were prepared by pyrolyzing the metal and carbon precursors (Co(NO_3_)_2_·6H_2_O and dicyandiamide) in N_2_ [11]. Typically, the mass ratio of Co(NO_3_)_2_·6H_2_O and dicyandiamide were kept at 1:10. A uniform pink paste was formed after adding suitable ethanol and continuously grinding in a crucible boat. Then, the paste was pyrolyzed at 800 °C with a heating rate of 3 °C·min^−1^ under N_2_ atmosphere. Dicyandiamide (C_2_H_4_N_4_) was used as a C and N source during the pyrolysis process. The Co^2+^ was reduced by dicyandiamide to form Co^0^, which would act as a catalyst to promote the formation of carbon nanotubes. After cooling down to room temperature, the products were leached in 0.5 M H_2_SO_4_ at 90 °C for 5 h to remove the undesirable impurities outside the carbon nanotubes, followed by being washed with PI water several times. Then, the catalysts were dried at 60 °C and characterized by X-ray diffraction (XRD, D8 advance, Bruker, Karlsruhe, Germany), scanning electron microscopy (FE-SEM, SU8200, HITACHI, Tokyo, Japan), X-ray photoelectron spectroscopy (XPS, ESCALAB 250Xi, Thermo Fisher Scientific, Waltham, MA, USA) and N_2_ sorption/desorption analysis (ASAP2020-HD88, Micromeritics, Atlanta, GA, USA).

### 2.2. The Preparation of Electrode and Electrocatalytic Production of H_2_O_2_

The preparation of the cathode for the electrocatalytic production of H_2_O_2_ was carried out as follows: 5 mg Co-N-C powders were well dispersed into a mixed solution of isopropanol (0.95 mL) and polytetrafluoroethylene suspension (PTFE, 0.05 mL, 60 wt%) [24]. Slurry-like specimens were formed by heating in an 80 °C water bath, and then, the resulting black slurry was uniformly coated on the nickel foam collector. The working electrode was then dried at 60 °C for 6 h and pressed into tablets for use. The schematic diagram of the electro-Fenton experimental process is shown in Figure 1.

The concentration of H_2_O_2_ in situ produced by the cathode under different conditions was tested using a spectrophotometer using the potassium titanium oxalate photometric method [25]. Then, 1.00 mL of a H_2_SO_4_ solution (3 M) and 1.00 mL of a K_2_TiO(C_2_O_4_)_2_ solution (0.05 M) were added into 2 mL of the tested solution. The mass concentration of H_2_O_2_ was obtained from the tested absorbance in the absorbance curve of the standard H_2_O_2_ solution [26].

### 2.3. Degradation Experiments

An E-Fenton system, using Co-N-C as a cathode and carbon as an anode, was constructed for the degradation of RhB (20 mg/L). A UV–vis spectrophotometer was used to measure the residual concentration of RhB in the solution. The initial pH of the RhB solution was about 6.5, and it was adjusted to 3 by H_2_SO_4_ (0.1 M) to promote E-Fenton efficiency and prevent the flocculation of iron ions. Typically, the RhB solution was pretreated with oxygen for 20 min, then followed by the E-Fenton degradation process. At the set time interval, solution samples of 0.9 mL were collected and filtered with a 0.45 μm membrane, and then quenched with MeOH (0.1 mL) at once [13].

## 3. Results and Discussion

### 3.1. Characterization of Co-N-C Composite Catalysts

As shown in Figure 2a, all three XRD patterns of the Co-N-CNTs synthesized at different temperatures included the diffraction peaks of the graphite (at 26.4°) and cobalt (at 44.2°, 51.5° and 75.6°, respectively), which perfectly matched with graphite (hexagonal C, PDF41-1487) and cubic Co (Co^0^, PDF: 15-0806) and indicated the presence of graphite and Co^0^.

The FTIR results in Figure 2b clearly reveal the characteristic peaks of -OH, -COO and aromatic C-H groups in all three Co-N-CNTs samples. The peaks at 2922, 1087 and 876 cm^−1^ were related to the aromatic C-H bonds, while the two peaks at 1631 and 1424 cm^−1^ were related to the stretching vibration of asymmetric ν_as_ (-COO) and symmetric ν_s_ (-COO). The FTIR spectra of Co-N-CNT showed that most of the organic carboxylate was removed after the pyrolysis process, which further verified the effective structure transformation from cobalt salts to Co-N-CNTs. The wide band at 3432 cm^−1^ was attributed to the stretching vibration of the -OH group. When the temperature increased from 700 °C to 900 °C, the intensity of the -OH group slightly decreased, which indicated that the acidity of the surface of the catalyst slightly decreased. In this case, the decreased surface acidity of the catalyst meant a hindered reduction activity of O_2_ to H_2_O_2_.

The pore distribution shown in Figure 2d proved the presence of a hybrid pore structure of a few micropores and the majority of mesopores with diameters centered at about 3 to 4 nm and BET specific surface areas of 142.89, 130.23 and 147.73 m^2^/g for Co-N-CNTs-700, Co-N-CNTs-800 and Co-N-CNTs-900, respectively. As shown in Figure 2c, the N_2_ adsorption–desorption isotherm of the Co-N-C was a type IV adsorption isotherm with a hysteresis loop at P/P_0_ from 0.4 to 0.5, which also indicated the irregular hybrid pore structure of micropores and mesopores in the Co-N-C.

As shown in Figure 3a–c, numerous carbon nanotubes were formed after pyrolysis for all three specimens of Co-N-CNTs synthesized at different temperatures. The walls of the nanotubes synthesized at 700 °C and 900 °C were rough and uneven, while the walls of those synthesized at 800 °C were smooth and uniform. The diameters of the nanotubes ranged from 40 to 400 nm, while the diameters of the nanotubes synthesized at 800 °C were more even. As shown in Figure 3d, cobalt nanoparticles encapsulated at the enclosed end of CNTs were also revealed by the images. The energy dispersive X-ray spectroscopy (EDS) elemental mapping of a selected area on the Co-N-C composite (Figure 3e–i) revealed the existence of cobalt nanoparticles at the enclosed end of CNTs and uniform distributions of the N, O and C elements across the tubular structure, which demonstrated the successful incorporation of Co, N and O into the carbon substrate. During the process of pyrolysis, the entrapped cobalt acted as a catalyst to promote the formation of carbon nanotubes [4,27].

As shown in Figure 4a, the X-ray photon spectroscopy (XPS) results also confirmed the existence of Co (0.80 at%), N (3.99 at%), O (3.40 at%) and C (91.81 at%) on the Co-N-C. The Co 2p XPS spectra of the freshly prepared Co-N-C also supported the existence of Co^0^ (peak at 778.5 eV), as well as Co-N (peak at 780.5 eV, Co^2+^ (peak at 782.8 eV) and Co^3+^ (peak at 795.5 eV), which was consistent with the XRD result (Figure 2a). The deconvolution results of C1s of the Co-N-C included four peaks attributed to C-C/C=C (at 284.8 eV), C-N (at 285.9 eV), C-O (at 286.8 eV) and N = C–N (at 287.5 eV). The N 1s of Co-N-C was deconvoluted into four peaks (402.52 ± 0.2 eV, 401.10 ± 0.2, 399.36 ± 0.2 and 398.64 ± 0.2, respectively, shown in Figure 4c), which corresponded to oxidized N, graphitic N, pyrrolic N and pyridinic N, respectively. Previous studies have reported that the introduction of doped nitrogen in CNTS could facilitate electron transport. The introduction of C–NHx into adsorbents could also boost the adsorptive removal of organics due to its outstanding capability for hydrogen bonding [26].

### 3.2. Electrochemical Performance Evaluation and Production Efficiency of H_2_O_2_

Figure 5 shows the CV, LSV and AC impedance spectroscopy curves of the Co-N-C electrode synthesized at 700 °C, 800 °C and 900 °C under oxygen saturation conditions. According to Figure 5, after oxygen permeation, all the three electrodes prepared by Co-N-C catalysts showed obvious O_2_ reductions at about −0.2 V to −0.3 V (vs. SCE), while there were no obvious O_2_ reductions under N_2_ saturation conditions. The LSV results were also consistent with the CV curves. From Figure 5c, we could see that semicircles appeared in the high-frequency region of the impedance diagram of all the three electrodes, indicating an electrochemically controlled step. In addition, the impedance arc radius of the Co-N-C electrode synthesized at 800 °C was the smallest, which indicated that this electrode had the smallest resistance (interface resistance of solution (R_s_ = 4.315 Ω) and contact resistance (R_CT_ = 11.05 Ω)) and the best electron transfer efficiency and electro-catalytic performance of oxygen reduction compared to other electrodes in the electrochemical reaction process. These results implied the Co-N-C electrode synthesized at 800 °C would have had the best EF degradation efficiency for RhB.

The yield of H_2_O_2_ was measured for the Co-N-C electrode synthesized at 800 °C under different applied voltages (as shown in Figure 6). There was little difference between the yield of H_2_O_2_ under different applied voltages, and the yield of H_2_O_2_ could reach 80 mg/L/h within 60 min, which was higher than that in many reported studies. From the literature, we know that a higher voltage can promote proton-coupled electron transfer in the formation of H_2_O_2_, while a side reaction of HER will also be promoted under higher voltages. Considering the influence of energy consumption and the HER side reaction, the optimal voltage for H_2_O_2_ yield in this system was −0.7 V (vs. SCE).

### 3.3. Efficient E-Fenton Degradation of RhB

The E-Fenton degradation efficiency of different systems for the removal of RhB was explored. Figure 7a shows that all the Co-N-C displayed efficient E-Fenton degradation rates of RhB, while that synthesized at 800 °C exhibited the highest RhB degradation (91% removal in 1 h). The reaction rate constants for the Co-N-C synthesized at 700 °C, 800 °C and 900 °C were 0.0353, 0.0366, and 0.0246, respectively. Figure 7c shows that all the Co-N-C displayed efficient E-Fenton degradation rates of RhB, while that synthesized at 800 °C exhibited the highest RhB degradation (91% removal in 1 h). The reaction rate constants for the Co-N-C system with different Fe^2+^ additions at 0.2 mM, 0.3 mM and 0.4 mM were 0.0228, 0.0366 and 0.0309, respectively. The highest RhB degradation rate was obtained with 0.3 mM Fe^2+^. During this process, the •OH could be generated from both the Co^2+^/Co^3+^ cycle and the Fe^2+^/Fe^3+^ cycle, which quickened the degradation of the RhB due to the Co-Fe synergy effect. An increase in Fe^2+^ concentration can promote the Fenton reaction to produce more ·OH, while an excess of Fe^2+^ will compete electrons with ·OH and reduce the contact of ·OH with organic compounds, which will lead to a decrease in the degradation efficiency of RhB.

A possible RhB degradation mechanism by the Co-N-C mediated E-Fenton system is illustrated in the following equations. In terms of the XPS results, Co-N-C contained Co^0^, Co^2+^ and Co^3+^. In addition to the Co ions, the metallic Co^0^ also played a role in activating E-Fenton. H_2_O_2_ and •OH could be generated from a Co^2+^/Co^3+^ redox cycle of the cobalt element during the E-Fenton process (Equations (1)–(4)), which ensured a high removal rate of RhB in this system [4]. The EPR results in Figure 7e reveal the existence of •OH, in which the four-line signal with an intensity ratio of 1:2:2:1 was attributed to DMPO-•OH. The active intermediate •OH could be generated from the self-formation process of E-Fenton. As a result, RhB molecules were degraded into intermediates and inorganic small molecules by these various ROSs involved in this system (Equations (5) and (6)) [28,29].
2H^+^ + O_2_ + 2e → H_2_O_2_
(1)
Co^2+^ + H_2_O_2_ + H^+^ → Co^3+^ + H_2_O + •OH (2)
Co + 2Co^3+^ → 3 Co^2+^
(3)
Co^3+^ + e → Co^2+^
(4)
•OH + RhB → intermediates (5)
intermediates + •OH → CO_2_ + H_2_O + Ions (6)

### 3.4. Reusability of Co-N-CNTS

The reusability of Co-N-CNTs was studied by fulfilling cycling tests. As shown in Figure 8, after three cycling runs, the degradation efficiency of RhB decreased from about 89.4% to 85.8%, and the degradation rate of RhB only decreased by about 4%, which proved that the Co-N-CNTs had very good reusability. The above result indicates that during the process of RhB degradation, the Co nanoparticles were tightly embedded in the carbon nanotube and drove a Co^2+^/Co^3+^ cycle to facilitate the E-Fenton degradation of RhB.

## 4. Conclusions

An E-Fenton oxidation system based on Co-N-CNTs with a hybrid pore structure of a few micropores and the majority of mesopores with diameters centered at about 3 to 4 nm and large BET specific surface areas was prepared, which enabled its good degradation efficiency for the degradation of RhB (91% at 60 min). The diameters of the Co-N-CNTs with cobalt nanoparticles encapsulated at the enclosed end ranged from 40 to 400 nm, and the yield of H_2_O_2_ could reach 80 mg/L/h at −0.7 V (vs. SCE). The Co-N-CNTs-800 had the smallest interface resistance of solution (R_s_ = 4.32 Ω) and contact resistance (R_CT_ = 11.05 Ω), which meant it had the best electron transfer efficiency and electro-catalytic performance of oxygen reduction. When the Fe^2+^ was added, the •OH could be generated from both the Co^2+^/Co^3+^ cycle and the Fe^2+^/Fe^3+^ cycle, which quickened the degradation of the RhB due to the Co-Fe synergy effect. After three cycling runs, 96% of the degradation efficiency of RhB could be retained. The core–shell structure (Co nanoparticles encapsulated in carbon nanotubes) could prevent Co from leaching, which enabled its good reusability and convenient recycling for its magnetism.

## Figures and Tables

**Figure 1 materials-16-02606-f001:**
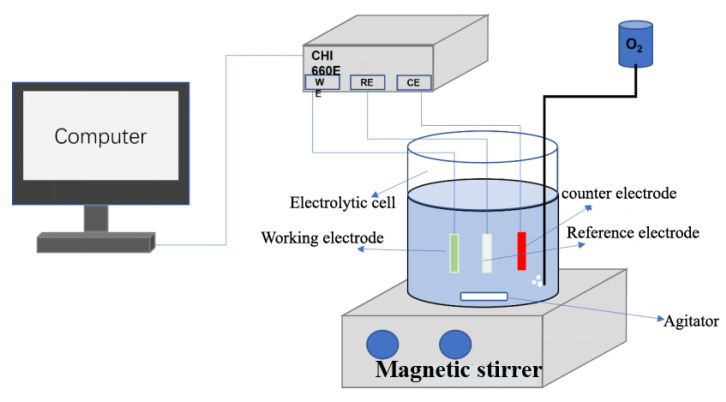
Schematic diagram of electro-Fenton experimental process.

**Figure 2 materials-16-02606-f002:**
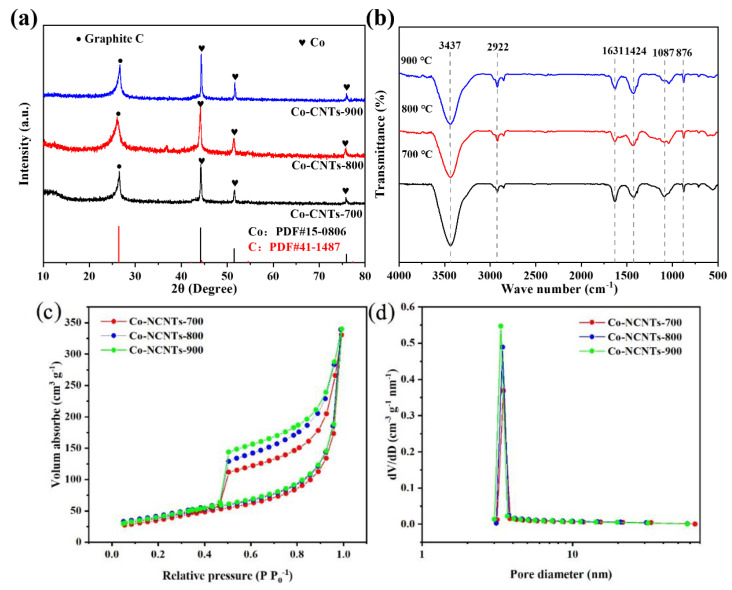
(**a**) XRD, (**b**) FTIR, (**c**) N_2_ adsorption–desorption isotherms and (**d**) pore size distribution patterns of Co−N−CNTs synthesized at different temperature.

**Figure 3 materials-16-02606-f003:**
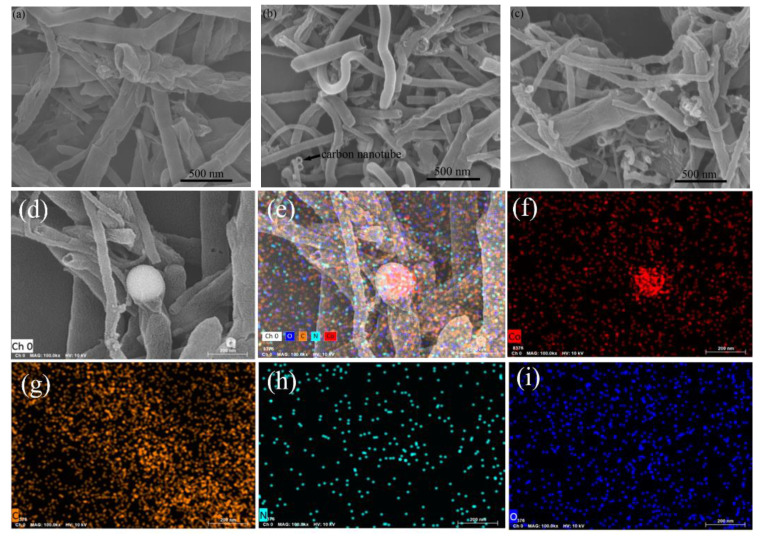
SEM images of Co−N−CNTs synthesized at different temperatures: (**a**) 700 °C, (**b**) 800 °C and (**c**) 900 °C; (**d**,**e**) EDS mapping images, (**f**) Co, (**g**) C, (**h**) N and (**i**) O of Co−N−CNTs at 800 °C.

**Figure 4 materials-16-02606-f004:**
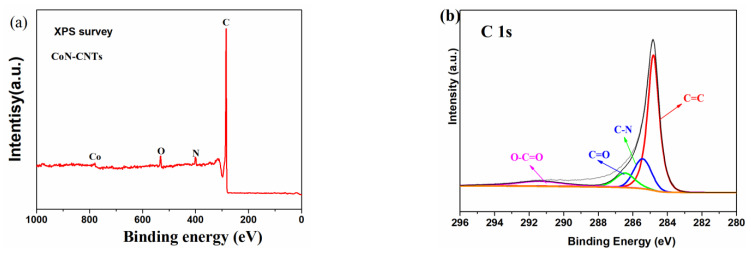
High-resolution XPS spectra of (**a**) XPS survey of the fresh and used Co-N-C, (**b**) C 1s, (**c**) N 1s, (**d**) Co 2p, and (**e**) O 1s.

**Figure 5 materials-16-02606-f005:**
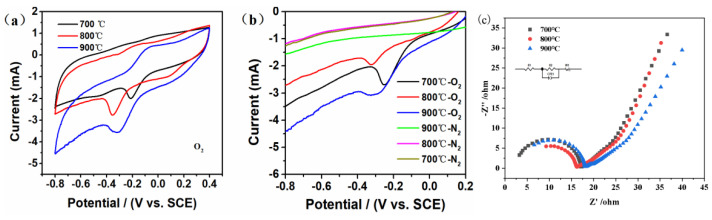
Electrochemical performance evaluation of the Co−N−C electrode synthesized at 700 °C, 800 °C and 900 °C under oxygen saturation condition. (**a**) CV, (**b**) LSV and (**c**) AC impedance spectroscopy.

**Figure 6 materials-16-02606-f006:**
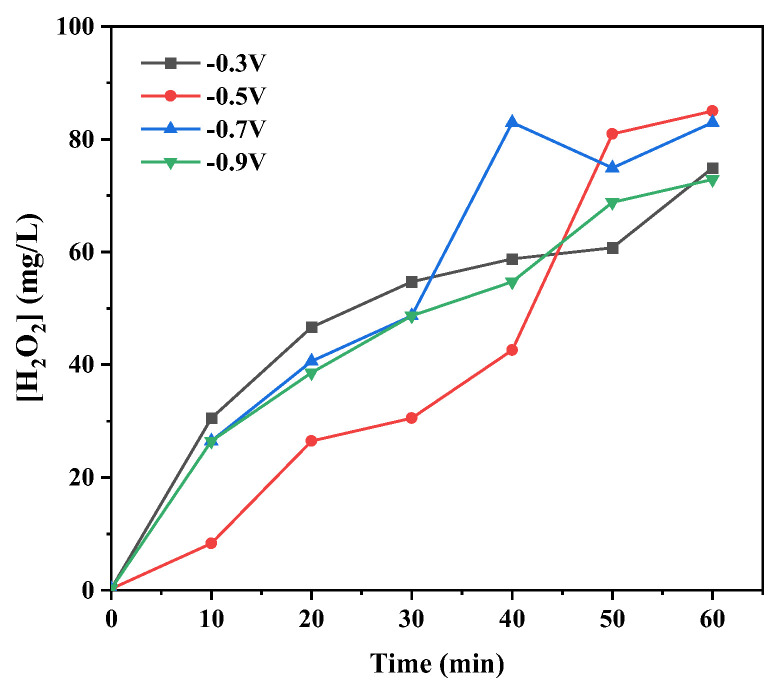
Electrochemical production efficiency of H_2_O_2_ for Co-N-C electrode synthesized at 800 °C under different applied voltages.

**Figure 7 materials-16-02606-f007:**
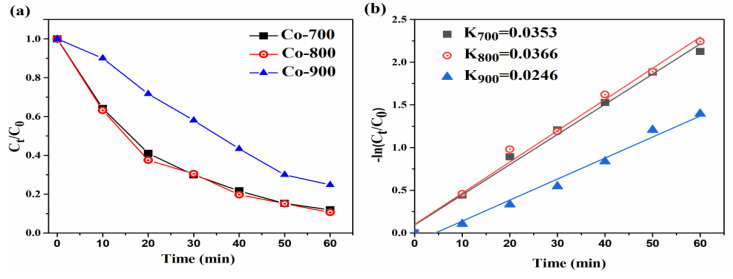
The E-Fenton degradation efficiency and reaction rate constants of (**a**,**b**) the Co-N-C synthesized at 700 °C, 800 °C and 900 °C, (**c**) and (**d**) different Fe^2+^ additions; (**e**) EPR results.

**Figure 8 materials-16-02606-f008:**
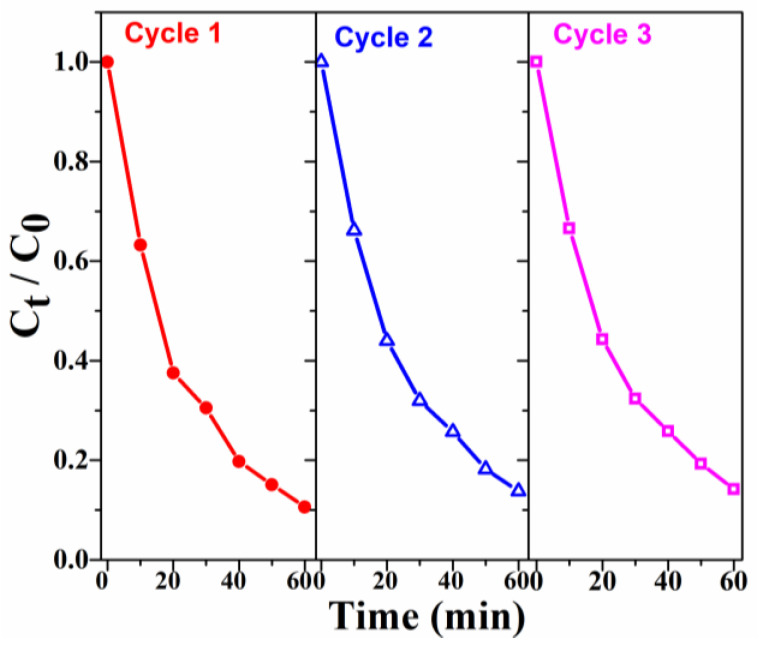
Cycling test of Co-N-CNTS for the E-Fenton degradation of RhB (initial RhB concentration =  20  mg L^−1^).

## Data Availability

The data supporting the findings of this paper are available from the corresponding author upon request.

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
