# Peer review of "Fast Degradation of Rhodamine B by In Situ H_2_O_2_ Fenton System with Co and N Co-Doped Carbon Nanotubes"

_materials, 2023, doi:10.3390/ma16072606_

Round 1
Reviewer 1 Report
Manuscript Number: materials-2267407
Title: “Fast Degradation of Rhodamine B by In-Situ H2O2 Fenton System with Co-N Carbon nanotubes”
Authors: Wei Cui, Jia hui Fang, Yuanyuan Wan, Xueyu Tao, Litong Guo and Qiyan Feng
The manuscript studies the degradation for Rhodamine B (RhB) in water, using an E-Fenton oxidation process based on Co-N co-doped carbon nanotubes (Co-N-CNTs). The authors prepared the Co-N co-doped carbon nanotubes with diameters from 40 to 400 nm and achieved a fast RhB removal of around 91% at 60 min. By increasing the Fe2+ concentrations from 0.3 to 0.4 mM, the RhB degradation efficiency decreased from 91% to about 87%.
I suggest publication of this manuscript after the authors consider the following comments and recommendations on a new version of the manuscript:
GENERAL COMMENT
For a general reader interested in this E-Fenton process, it would be convenient to give more details about the experimental setup to carry out the experimental runs. For example, was a device used to keep the working temperature constant? Was a cover used over the reactor surface to prevent loss of volatile reactants? Perhaps, a simple scheme or representation of the experimental setup can be helpful.
Page 1, last line:
In the Introduction, the authors announce that "The Co based catalysts embedded in carbon matrix which have a large specific surface area can effectively improve the catalytic activity and prevent Co from leaching [15,8]". Was the presence of small amounts of Co analysed in the final aqueous solution? Have they carried out repetitive tests of RhB degradation to show that the synthesized catalyst remains practically unchanged over multiple tests? They must clarify this point.
Page 2, Section 2.2, line 28:
“PTFE” meaning should be declared the first time it is mentioned in the manuscript.
Page 2, Section 2.3, lines 37-39:
Here, the authors state that "An E-fenton system by using Co-N-C as cathode and carbon as anode was constructed for the degradation of RhB (20mg/L). The UV–vis spectrophotometer was used to measure the residual concentration of RhB in solution".
However, it is well known that some intermediate products could be formed during the Rhodamine B degradation. Could these undetected intermediate compounds be toxic? The authors should demonstrate that the organic compounds remaining at the end of the treatment are non-toxic compounds.
Page 2, Section 2.3, lines 39-40:
In this section, the authors affirm that "The pH of the RhB solution was adjusted by H2SO4 (0.1 M) or NaOH (0.1 M) to avoid the influence of introducing Cl-". However, they do not declare the working pH value of the experiments. They must provide this pH value.
Figure 6(c) and (d):
In Figures 6(c) and (d) the authors have shown tests with Fe(2+) concentrations up to 0.4 mM. Bearing in mind that these water treatments are usually carried out to throw them into rivers or lakes, what is the maximum legal (or acceptable) concentration of Fe in mg/L to throw into water sources in the region? They must clarify this point.
Page 7, Equations (1) to (6):
In Equations (1) to (6) the authors propose a possible mechanism of RhB degradation together with the generation of intermediates and inorganic small molecules. However, as explained above, intermediate products can be formed which could be toxic. Therefore, the authors have to measure the global contamination at the end of the treatment, for example: Total Organic Carbon, Chemical Oxygen Demand, Toxicity, etc.
Page 7, Equation (1):
Note that in Equation (1) of the RhB degradation mechanism, the positive charges on both sides of the equation are not balanced.
MINOR COMMENTS
Page 3, line 3:
“The pore distribution shown in Fig. 1 (b)…” should be replaced by “The pore distribution shown in Fig. 1 (c)…”.
Page 3, line 5:
“As shown in Fig. 1 (c), the N2 adsorption–desorption isotherms…” should be replaced by “As shown in Fig. 1 (b), the N2 adsorption–desorption isotherms…”.
Author Response
Reviewer #1: The manuscript studies the degradation for Rhodamine B (RhB) in water, using an E-Fenton oxidation process based on Co-N co-doped carbon nanotubes (Co-N-CNTs). The authors prepared the Co-N co-doped carbon nanotubes with diameters from 40 to 400 nm and achieved a fast RhB removal of around 91% at 60 min. By increasing the Fe2+ concentrations from 0.3 to 0.4 mM, the RhB degradation efficiency decreased from 91% to about 87%.
I suggest publication of this manuscript after the authors consider the following comments and recommendations on a new version of the manuscript:
For a general reader interested in this E-Fenton process, it would be convenient to give more details about the experimental setup to carry out the experimental runs. For example, was a device used to keep the working temperature constant? Was a cover used over the reactor surface to prevent loss of volatile reactants? Perhaps, a simple scheme or representation of the experimental setup can be helpful.
Answer: A simple scheme of the experimental setup was added according to the reviewer’s comment. Some details about the experimental setup were also added.
- Page 1, last line:In the Introduction, the authors announce that "The Co based catalysts embedded in carbon matrix which have a large specific surface area can effectively improve the catalytic activity and prevent Co from leaching [15,8]". Was the presence of small amounts of Co analysed in the final aqueous solution? Have they carried out repetitive tests of RhB degradation to show that the synthesized catalyst remains practically unchanged over multiple tests? They must clarify this point.
Answer: The repetitive tests of RhB degradation was added according to the reviewer’s comment. We have not analysed the presence of small amounts of Co in the final aqueous solution. But we do analyse the content of Co in the Used and Fresh prepared Co-N-CNTS. The content of Co remained almost the same (total Co elment 0.80 at% vs 0.78 at%, while Co0 0.37at% vs 0.36at%).
- Page 2, Section 2.2, line 28:“PTFE” meaning should be declared the first time it is mentioned in the manuscript.
Answer: The meaning of “PTFE” have be added according to the reviewer’s comment.
- Page 2, Section 2.3, lines 37-39:Here, the authors state that "An E-fenton system by using Co-N-C as cathode and carbon as anode was constructed for the degradation of RhB (20mg/L). The UV–vis spectrophotometer was used to measure the residual concentration of RhB in solution".
However, it is well known that some intermediate products could be formed during the Rhodamine B degradation. Could these undetected intermediate compounds be toxic? The authors should demonstrate that the organic compounds remaining at the end of the treatment are non-toxic compounds.
Answer: Due to the time, the intermediate compounds have not been studied. But we have test the TOC values of the original RhB solution and the RhB solution after E-fenton process. There were about 41.43% of the TOC values remained in the solution (7.396 mg/L VS 17.85mg/L), which indicated that about 59.57% of the original RhB was transformed to inorganic carbon(carbon dioxide). Because there were also about 10% of the RhB was remained in the solution, about 30% of the intermediate compounds was formed during the E-Fenton process.
- Page 2, Section 2.3, lines 39-40:
In this section, the authors affirm that "The pH of the RhB solution was adjusted by H2SO4 (0.1 M) or NaOH (0.1 M) to avoid the influence of introducing Cl-". However, they do not declare the working pH value of the experiments. They must provide this pH value.
Answer: The working pH value of the experiments have be added according to the reviewer’s comment.
- Figure 6(c) and (d):
In Figures 6(c) and (d) the authors have shown tests with Fe(2+) concentrations up to 0.4 mM. Bearing in mind that these water treatments are usually carried out to throw them into rivers or lakes, what is the maximum legal (or acceptable) concentration of Fe in mg/L to throw into water sources in the region? They must clarify this point.
Answer: The Fe2+ was added into the solution to drive a Fe2+/Fe3+ cycle for accelerating the E-fenton process. After the removal of the organic pollutants, it can be removed by flocculation.
- Page 7, Equations (1) to (6):
In Equations (1) to (6) the authors propose a possible mechanism of RhB degradation together with the generation of intermediates and inorganic small molecules. However, as explained above, intermediate products can be formed which could be toxic. Therefore, the authors have to measure the global contamination at the end of the treatment, for example: Total Organic Carbon, Chemical Oxygen Demand, Toxicity, etc.
Answer: The Total Organic Carbon have be tested according to the reviewer’s comment. We have test the TOC values of the original RhB solution and the RhB solution after E-fenton process. There were about 41.43% of the TOC values remained in the solution (7.396 mg/L VS 17.85mg/L), which indicated that about 59.57% of the original RhB was transformed to inorganic carbon(carbon dioxide). Because there were also about 10% of the RhB was remained in the solution, about 30% of the intermediate compounds was formed during the E-Fenton process.
- Page 7, Equation (1):
Note that in Equation (1) of the RhB degradation mechanism, the positive charges on both sides of the equation are not balanced.
Answer: The Equation (1) has be revised according to the reviewer’s comment.
- MINOR COMMENTS
Page 3, line 3:
“The pore distribution shown in Fig. 1 (b)…” should be replaced by “The pore distribution shown in Fig. 1 (c)…”.
Page 3, line 5:
“As shown in Fig. 1 (c), the N2 adsorption–desorption isotherms…” should be replaced by “As shown in Fig. 1 (b), the N2 adsorption–desorption isotherms…”
Answer: The mistake has been revised according to the reviewer’s comment.

Reviewer 2 Report
See the attached document

Author Response
Our manuscript, referenced above, has been revised according to the referee’s comments. Your comments for our manuscript are very thoughtful and valuable for revising our paper and improving our work, we appreciate it very much. A detailed answer to all the queries has been provided. I list the modifications as follows:
Reviewer #2: The manuscript reports the design of electrochemical Fenton system based on Cobalt and Nitrogen codoped carbon nanotubes composite catalyst serving as a cathode in the experiments studying the electrochemical degradation of Rhodamine B by the generated hydroxyl radicals. The results demonstrate the efficiency of the designed catalytical system and are worth publishing after the manuscript undergoes substantial revisions for clarity of presentation to a broader readership.
Relative comments are listed here:
- The revised title of the manuscript is recommended to define more accurately the system under study, such as “…System with Co and N co-doped Carbon Nanotubes”.
Answer: The title has been updated according to the reviewer’s comment.
- The system under study (according to the title) is Co-N-CNT, however, in many sections of the manuscript it is also abbreviated as Co-N-C. This discrepancy should be addressed.
Answer: The discrepancy has been revised according to the reviewer’s comment.
- Method for synthesis of carbon nanotubes is not clearly described in the Experimental Section. It only mentions “carbon precursors”. So, how CNTs are prepared? What is the chemical path for their formation from”carbon precursor”?
Answer: The method for synthesis of carbon nanotubes has been revised according to the reviewer’s comment. Typically, the mass ratio of Co(NO3)2·6H2O and dicyandiamide were kept at 1:10. The dicyandiamide (C2H4N4) was used as C and N sources during the pyrolysis process. The Co2+ was reduced by dicyandiamide to form Co nano-particles, which will act as a catalyst to promote the formation of carbon nanotubes.
- The sentence “The preparation of cathode for electrocatalytic production of H2O2 were as follows: 5 mg Co-N-C powders were dispersed in 0.95 mL isopropanol solution and 0.05 mL 60 wt% PTFE solution [17].” is hard to understand.
Answer: This sentence has been revised according to the reviewer’s comment.
- The same applies to “Then 1 mL solution of H2SO4 (3 M) and K2TiO(C2O4)2 (0.05 M) were added into 2 mL water solution, successively.” (?)
Answer: This section has been revised according to the reviewer’s comment.
- In the section “The pore distribution shown in Fig. 1 (b) ,,,,,,mesopores in the Co-N-C.” figures (b) and (c) are confused.
Answer: This section was revised according to the reviewer’s comment.
- What kind of chemical structure is “DMPO-•OH.” ? Please explain.
Answer: The DMPO is used as a radical trap for •OH. Here, we use “DMPO-•” to present the radical adducts formed between DMPO and •OH.
- Chemical equations 1-3 are not presented in the acceptable way.
Answer: Chemical equations 1-3 have been revised according to the reviewer’s comment.
- Conclusions entirely duplicate the Abstract. This is not acceptable too
Answer: Conclusions have been updated according to the reviewer’s comment.
Thanks for the reviewer’s suggestions to us. It is very helpful to our manuscript.

Reviewer 3 Report
Manuscript ID: materials-2267407
Wei Cui, Jia hui Fangand, co-authors reported" Fast Degradation of Rhodamine B by In-Situ H2O2 Fenton System with Co-N Carbon nanotubes". Although the topic is interesting, but some important aspects were not performed. Following comments should be addressed before possible consideration for publication in worthy Journal of Materials. I believe it will not take a long for the authors to work on this revision. My comments are,
1. Abstract should be consist of some characterization findings like SEM & XRD.
2. There are so many typo grammatical errors in whole manuscript, should be revised by some native speaker and formatting should be checked
3. More key words like Rhodamine B should be added
4. Novelty of work should be described in introduction section
5. % purity of materials used should be specified.
6. Presentation of Figs not corrected all over the manuscript.
7. Mechanism of adsorption should be discussed graphically.
8. Disposed Dyes like Rhodamine B of different industries should be discussed with their side effects for reference, Surfaces and Interfaces 34 (2022) 102324.
9. In introduction more literature should be reviewed and some latest adsorbents should be discussed here to enhance the novelty of work like, Optical Materials 126 (2022) 112199, Chemical Physics Letters 805 (2022) 139939
10. FTIR analysis should be carried out to check the structural changes of Co-N-CNTs synthesized at different temperature.
11. Introduction is too short it should be include more literature about synthesis methods of Co-N-CNTs.
12. As authors claimed that the synthesized material is microporous. Then how you confirmed that this is photodegradation not adsorption? justify
13. Reusability and stability test should be performed
14. In conclusion more data should be summarized
15. References are too short; it’s not enough for paper and to explain their relatable material should be cite more related and advanced work.
Author Response
Reviewer 3#: Wei Cui, Jia hui Fangand, co-authors reported" Fast Degradation of Rhodamine B by In-Situ H2O2 Fenton System with Co-N Carbon nanotubes". Although the topic is interesting, but some important aspects were not performed. Following comments should be addressed before possible consideration for publication in worthy Journal of Materials. I believe it will not take a long for the authors to work on this revision. My comments are,
- Abstract should be consist of some characterization findings like SEM & XRD.
Answer: The abstract has been revised according to the reviewer’s comment.
- There are so many typo grammatical errors in whole manuscript, should be revised by some native speaker and formatting should be checked
Answer: The typo grammatical errors have been revised according to the reviewer’s comment.
- More key words like Rhodamine B should be added.
Answer: The key words have been revised according to the reviewer’s comment.
- Novelty of work should be described in introduction section
Answer: The novelty of work has been added according to the reviewer’s comment.
- % purity of materials used should be specified.
Answer: The purity of materials used has been specified according to the reviewer’s comment.
- Presentation of Figs not corrected all over the manuscript.
Answer: The presentation of Figs has been revised according to the reviewer’s comment.
- Mechanism of adsorption should be discussed graphically.
Answer: We thanks for the reviewer’s suggestion. We have investigated the adsorption performance of the Co-N-CNTS synthesized in this paper, as shown in the figure below. From this figure, we can see that the adsorption was only about 10%, while the degradation could reach at about 90% during 1h E-fenton process. Because the adsorption was only about 10%, so we did not add an discussion about the mechanism of adsorption.
- Disposed Dyes like Rhodamine B of different industries should be discussed with their side effects for reference, Surfaces and Interfaces 34 (2022) 102324.
Answer: We thanks for the reviewer’s suggestion. Due to the time, we have not studied the composition of the intermediate compounds. So, the side effects have not been discussed here. But we have added this reference into our manuscript to direct our study in this research.
- In introduction more literature should be reviewed and some latest adsorbents should be discussed here to enhance the novelty of work like, Optical Materials 126 (2022) 112199, Chemical Physics Letters 805 (2022) 139939
Answer: We thanks for the reviewer’s suggestion. More literature have been added into the introduction to enhance the novelty of work according to the reviewer’s comment.
- FTIR analysis should be carried out to check the structural changes of Co-N-CNTs synthesized at different temperature.
Answer: FTIR analysis has been be carried out to check the structural changes of Co-N-CNTs according to the reviewer’s comment.
- Introduction is too short it should be include more literature about synthesis methods of Co-N-CNTs.
Answer: Introduction has been be revised according to the reviewer’s comment.
- As authors claimed that the synthesized material is microporous. Then how you confirmed that this is photodegradation not adsorption? justify
Answer: We have investigated the adsorption performance of the Co-N-CNTS synthesized in this paper, as shown in the figure below. From this figure, we can see that the adsorption was only about 10%, while the degradation could reach at about 90% during 1h E-fenton process.
- Reusability and stability test should be performed
Answer: Introduction has been be revised according to the reviewer’s comment.
- In conclusion more data should be summarized
Answer: The conclusion section has been be revised according to the reviewer’s comment.
- References are too short; it’s not enough for paper and to explain their relatable material should be cite more related and advanced work.
Answer: References has been be revised according to the reviewer’s comment.
Round 2
Reviewer 1 Report
Manuscript Number: materials-2267407-R1
Title: “Fast Degradation of Rhodamine B by In-Situ H2O2 Fenton System with Co- and N co-doped Carbon Nanotubes”
Authors: Wei Cui, Jia hui Fang, Yuanyuan Wan, Xueyu Tao, Litong Guo and Qiyan Feng
In the reviewed manuscript, the authors have adequately taken into account most of my comments, with the exception of the following comment No. 5:
Page 2, Section 2.3, lines 39-40:
In this section, the authors affirm that "The pH of the RhB solution was adjusted by H2SO4 (0.1 M) or NaOH (0.1 M) to avoid the influence of introducing Cl-". However, they do not declare the working pH value of the experiments. They must provide this pH value.
I have failed to find the working pH value in the reviewed manuscript, as requested in my comment No 5. The authors have not provided the page and line where it was added.
Author Response
In the reviewed manuscript, the authors have adequately taken into account most of my comments, with the exception of the following comment No. 5:
Page 2, Section 2.3, lines 39-40:
In this section, the authors affirm that "The pH of the RhB solution was adjusted by H2SO4 (0.1 M) or NaOH (0.1 M) to avoid the influence of introducing Cl-". However, they do not declare the working pH value of the experiments. They must provide this pH value.
I have failed to find the working pH value in the reviewed manuscript, as requested in my comment No 5. The authors have not provided the page and line where it was added.
Answer: The working pH value of the experiments have be added according to the reviewer’s comment. We are so sorry that we forgot to add this into the last revised manuscript.
Reviewer 3 Report
Accept in present form
Author Response
Comments and Suggestions for Authors
Accept in present form
Thanks for the reviewer’s suggestions to us. It is very helpful to our manuscript.
Your comments for our manuscript are very thoughtful and valuable for revising our paper and improving our work